# Synergistic Antifungal Effect and In Vivo Toxicity of a Monoterpene Isoespintanol Obtained from *Oxandra xylopioides* Diels

**DOI:** 10.3390/molecules29184417

**Published:** 2024-09-17

**Authors:** Orfa Inés Contreras-Martínez, Alberto Angulo-Ortíz, Gilmar Santafé Patiño, Jesus Sierra Martinez, Ricardo Berrio Soto, Joice Margareth de Almeida Rodolpho, Krissia Franco de Godoy, Fernanda de Freitas Aníbal, Bruna Dias de Lima Fragelli

**Affiliations:** 1Biology Department, Faculty of Basic Sciences, University of Córdoba, Montería 230002, Colombia; oicontreras@correo.unicordoba.edu.co (O.I.C.-M.); rberriosoto54@correo.unicordoba.edu.co (R.B.S.); 2Chemistry Department, Faculty of Basic Sciences, University of Córdoba, Montería 230002, Colombia; gsantafe@correo.unicordoba.edu.co; 3Department of Genetics and Evolution, Federal University of São Carlos, São Carlos 13565-905, SP, Brazil; jesussierra@estudante.ufscar.br; 4Laboratory of Inflammation and Infectious Diseases, Department of Morphology and Pathology, Federal University of São Carlos, São Carlos 13565-905, SP, Brazil; j_jrodolpho@hotmail.com (J.M.d.A.R.); krissia.fgodoy@gmail.com (K.F.d.G.); ffanibal@ufscar.br (F.d.F.A.); 5Functional Materials Development Center, Interdisciplinary Laboratory of Electrochemistry and Ceramics, Department of Chemistry, Federal University of São Carlos, São Carlos 13565-905, SP, Brazil; brufragelli@gmail.com

**Keywords:** synergism, antifungal, isoespintanol, in vivo toxicity

## Abstract

*Candida* sp. infections are a threat to global health, with high morbidity and mortality rates due to drug resistance, especially in immunocompromised people. For this reason, the search for new alternatives is urgent, and in recent years, a combined therapy with natural compounds has been proposed. Considering the biological potential of isoespintanol (ISO) and continuing its study, the objective of this research was to assess the effect of ISO in combination with the antifungals fluconazole (FLZ), amphotericin B (AFB) and caspofungin (CASP) against clinical isolates of *C. tropicalis* and to evaluate the cytotoxic effect of this compound in the acute phase (days 0 and 14) and chronic phase (days 0, 14, 28, 42, 56, 70 and 84) in female mice (*Mus musculus*) of the Balb/c lineage. The results show that ISO can potentiate the effect of FLZ, AFB and CASP, showing synergism with these antifungals. An evaluation of the mice via direct observation showed no behavioral changes or variations in weight during treatment; furthermore, an analysis of the cytokines IFN-γ and TNF in plasma, peritoneal cavity lavage (PCL) and bronchoalveolar lavage (BAL) indicated that there was no inflammation process. In addition, histopathological studies of the lungs, liver and kidneys showed no signs of toxicity caused by ISO. This was consistent with an analysis of oxaloacetic transaminases (GOT) and pyruvic transaminases (GPT), which remained in the standard range. These findings indicate that ISO does not have a cytotoxic effect at the doses evaluated, placing it as a monoterpene of interest in the search for compounds with pharmacological potential.

## 1. Introduction

The adaptive capacity of *Candida* spp. in hospital environments—together with the expression of various virulence factors, the poor availability of antifungal drugs and expressed drug resistance—represents a significant challenge in medical practice. Natural compounds, especially those from plants, play a primary role as a source of specialized metabolites with recognized medicinal properties. Owing to their wide chemical diversity, these metabolites can be directly used as bioactive compounds, drug prototypes or pharmacological tools for different targets [1,2,3]. Furthermore, potential efficacy and minimal side effects are the key advantages of plant-based products, making them appropriate tools in therapeutic treatments [4]. They are also an inspiration for synthesizing non-natural molecules and provide important clues for identifying and developing synergistic drugs [5].

Combination therapy is an interesting area in the development of new therapeutic strategies against fungal infections, and its advantages include the following: (i) a potential increase in effectiveness and the degree of elimination for these pathogens; (ii) a broader spectrum of activity directed at potentially resistant pathogens; (iii) preventing the emergence of resistance; and (iv) potentially reduced doses of individual antifungal drugs that may minimize toxicity [6,7]. The great efficacy of antifungal drugs such as fluconazole (FLZ), amphotericin B (AFB), nystatin and caspofungin (CASP) in combination with monoterpene phenols against *Candida auris* has been demonstrated [8]. Furthermore, the synergism of *Ocimum sanctum* essential oils with FLZ and ketoconazole (KZL) against different candida isolates, including both susceptible and resistant strains [9], has also been reported; likewise, the potentiation of AFB’s antifungal action through the essential oil of *Thymus vulgaris* has been indicated [10]. In monoterpenes such as thymol, a synergistic effect has been demonstrated in combination with antifungals such as nystatin against *Candida* spp. in oral cavity infection [11]; likewise, thymol and carvacrol have been documented with synergistic action with FLZ [12,13]. The combined therapy of thymol and carvacrol with caprylic acid (CA) and its mechanism of action against *Candida* spp. has also been documented [13]. Thymol has also shown synergy with FLZ against the planktonic growth mode and biofilms of resistant *C. albicans* and *C. tropicalis* strains; however, its synergy with AFB has only been evidenced in the planktonic cells of Candida [14]. Other interaction studies on azoles with *Mentha piperita* essential oils have shown synergistic profiles against *Candida* spp., *Cryptococcus neoformans* and *Trichophyton mentagrophytes* [15].

Isoespintanol (ISO) (2-isopropyl-3,6-dimethoxy-5-methylphenol) is a monoterpene isolated from *Oxandra xylopioides* (Annonaceae) that is recognized as a natural bioactive compound. Its biological potential has been reported as an antioxidant [16], anti-inflammatory [17], antispasmodic [18], vasodilator [19], cardioprotective [20] and cryoprotectant in canine semen [21]. Using this monoterpene as a therapeutic tool to prevent diabetes in the early stages of its development has also been suggested [22]. In previous works, we reported the potential of this monoterpene against human pathogens, specifically, hospital-acquired bacteria [23] and the yeasts of the Candida genus, demonstrating its action against different cellular targets and its ability to reprogram the transcriptome of these yeasts [24,25,26,27], as well as its cytotoxic potential against human tumor cell lines [28].

Considering the important biological potential of ISO, including its antifungal effect against FLZ-resistant yeasts and important differences between in vitro and in vivo experimental environments [28], we hypothesize that it can potentiate the effect of commercial antifungals; furthermore, it has no significant cytotoxic effect in vivo at low concentrations. This research evaluates the effect of ISO in combination with the antifungals FLZ, AFB and CASP against clinical isolates of *C. tropicalis* and the cytotoxic effect of this compound in female mice (*Mus musculus*). This will contribute to the study of this monoterpene, which could serve as an important tool in treating and controlling infectious diseases.

## 2. Results

### 2.1. Obtaining and Identification of Isoespintanol

ISO was obtained as a crystalline amorphous solid from a petroleum benzine extract of *O. xylopioides* leaves, and its structural identification was performed via GC-MS, ^1^H-NMR, ^13^C-NMR, DEPT, COSY ^1^H-^1^H, HMQC and HMBC. Information related to obtaining and identifying this ISO was reported in our previous study [25].

### 2.2. Isoespintanol Action in Combination with Commercial Antifungals

Figure 1 shows that the ISO-FLZ combination had a synergistic effect against six of the seven (107, 98, 92, 81, 74 and 03) *C. tropicalis* isolates evaluated; conversely, the ISO-AFB treatment showed synergistic action against two isolates (107 and 74), and the ISO-CASP treatment only showed a synergistic effect against one of the isolates (74). The antagonistic activity of ISO-AFB (isolate 98) and ISO-CASP (isolates 92 and 84) was also evidenced. Table 1 shows that the minimum inhibitory concentrations at which 90% of the yeasts were inhibited (MIC_90_) in synergistic activity were substantially lower than the individual MIC_90_ value of the ISO and antifungals used. The MIC_90_ values decreased by up to 138.5 and 2 times in FLZ, AFB and CASP.

Table 1 shows the individual and combined MIC_90_ values of ISO, FLZ, AFB and CASP, as well as the FIC indices and the observed effect.

### 2.3. ISO In Vivo Toxicity Experiments

#### 2.3.1. Animal Weight

Table 2 and Table 3 show that during the acute and chronic phase experiments, respectively, the average weight in grams of the animals during the 14 days (acute phase) and 84 days (chronic phase) did not decrease; there was also no mortality among the mice with any of the ISO doses evaluated in these experiments.

#### 2.3.2. Global Cell Count

Figure 2 shows the global profile of leukocytes in the blood, peritoneal cavity lavage (PCL) and bronchoalveolar lavage (BAL) during the acute phase (Figure 2A–C) and chronic phase (Figure 2D–F), referring to both the control group and the groups inoculated with ISO. A significant reduction was observed in inoculated groups G3, G4 and G6 compared with G1 (CTRL + water) and in G4 and G6 compared with G2 (Figure 2A). When counting leukocytes in the PCL and BAL, the values of G3, G4, G5 and G6 compared with groups G1 and G2 did not show significant differences in cell reduction (Figure 2B,C); however, there was a significant increase in G5 compared with G2 in the BAL (Figure 2C). In the chronic phase, global blood counts, PCL and BAL did not significantly differ in any group compared with groups G1 and G2 (Figure 2D–F). Finally, the overall cell count was not significantly affected by the ISO treatment.

#### 2.3.3. Differential Cell Counting

Figure 3 shows the differential count of the blood leukocytes, PCL and BAL in the acute phase for both the control group and the groups inoculated with ISO. There were significant differences in the number of monocytes in the blood in G3, G4 and G6 compared with G1 and G2 (Figure 3D). Likewise, significant differences were found in the number of neutrophils in the blood in G4 compared with G1 (Figure 3E). There was also an increase with a significant difference in the number of eosinophils in the blood in G3, G4, G5 and G6 compared with G1 and G2 (Figure 3F).

Figure 4 shows the differential counts of leukocytes in the blood, PCL and BAL in the chronic phase for both the control group and the groups inoculated with ISO. In the PCL, there was a significant decrease in monocytes in G5 compared with G1 and G2 (Figure 4A). In the BAL, neutrophils increased in G3 and G5 compared with group G2, and there was a significant increase in these same cells for G4 compared with G1 and G2. In the blood, a significant increase in neutrophils was evident in G3 and G5 compared with group G1. Eosinophils significantly increased in G3, G4, G5 and G6 compared with G1 and G2.

#### 2.3.4. GOT and GPT Dosage

Figure 5 shows the doses of hepatic transaminases, oxaloacetic transaminase (GOT) and pyruvic transaminase (GPT) in the plasma of the animals after inoculation with ISO in the acute (Figure 5A,B) and chronic phases (Figure 5C). There was no increase or significant difference in the amounts of GOT or GPT in the groups exposed to the compound compared with the control group in the acute phase. In the chronic phase, GPT levels significantly decreased in G4 and G5 compared with G2 (Figure 5C). There was no GOT expression.

#### 2.3.5. Measurement of Cytokines IFN-γ and TNF

The concentrations of IFN-γ and TNF (in pg/mL) were only detected in the plasma, BAL and PCL of each animal in blood in both the acute and chronic phases, as shown in Figure 6. In the acute phase (Figure 6A,B), no statistically significant values were detected when G3, G4, G5 and G6 were compared with controls G1 and G2. In the chronic phase (Figure 6C), no positive cytokine levels (in pg/mL) were detected.

#### 2.3.6. Histopathological Analysis of Organs (Lung, Liver and Kidney)

The renal fragments presented few microscopic alterations (based on discrete inflammatory infiltrates, mainly in cortical areas) (Figure 7); the lung fragments showed few changes, including hemorrhage, inflammatory infiltrate and thickening of the alveolar septa (Figure 8). Most of the liver fragments presented perivascular infiltrates, and pyogranulomatous granulomas were identified in only two cases (in the oil control and the chronic study with 200 mg) (Figure 9). However, none of the histopathological changes that occurred during the ISO treatment were significant or considered toxic at the doses evaluated. For details on the histopathological analysis, see the Appendix A.

## 3. Discussion

The increasing incidence of fungal infections, particularly in immunocompromised people, continues to be a serious public health problem globally. *Candida* spp. are the pathogens mainly involved; the wide range of pathogenicity factors expressed, the ability to respond efficiently to environmental stress, resistance to antifungal agents and the high mortality reported make medical practice a great challenge [29,30,31]. Thus, searching for alternatives to control and treat these pathogens is urgent. In this context, the advantages of combined therapy are an interesting alternative [6,7,15,32].

Here, we report the synergistic effect of ISO in combination with the antifungals FLZ, AFB and CASP, highlighting the effect achieved with ISO-FLZ against clinical isolates of *C. tropicalis* resistant to FLZ. We also report the synergistic effect of ISO-AFB and ISO-CASP, although not in all cases. These results are consistent with the studies that show the synergistic effect of the monoterpene thymol in combination with FLZ against azole-resistant *C. tropicalis*. As a consequence of these synergistic interactions, the mitochondrial membrane potential was reduced, the production of mitochondrial superoxide increased and alterations were observed in the nuclear morphology, surface and cell ultrastructure [33]. Other studies have shown the excellent synergistic activity of monoterpenes such as linalool, citral and citronellal combined with FLZ against strains of *C. albicans* resistant to FLZ [34]. Likewise, the greater efficacy of antifungal drugs in combination with monoterpene phenols has been reported against *C. auris* [8].

Combined monoterpene and antifungal therapy has been widely documented, including its important synergistic effect against *Candida* spp. The mechanisms involved in this synergistic activity include inhibiting different stages in the fungal intracellular pathways essential for survival, increasing the penetration of one antifungal agent due to the action of another on the fungal cell membrane, inhibiting transporter proteins and simultaneously inhibiting different cellular targets [11,35]. Monoterpenes such as thymol and carvacrol in combination with FLZ have shown a synergistic effect against *Candida* spp. [12]; combined thymol and nystatin therapy against *Candida* spp. has also been documented [11]. Using combined thymol–FLZ treatments against biofilms and the planktonic growth of resistant *C. albicans* and *C. tropicalis* strains has also been reported, indicating synergistic action; likewise, thymol–AFB has demonstrated synergistic action in the planktonic growth phase of *Candida* spp. [14]. Studies have reported mixed thymol–carvacrol and caprylic acid (CA) treatments against *Candida* spp., revealing synergism and indicating that their mechanism may involve membrane damage caused by CA, facilitating the entry of antifungal agents into the cytoplasm and the inhibition of efflux pumps by CA, carvacrol or thymol. This causes them to accumulate inside cells, causing cell death [13]. Combined essential oils and antifungal therapies such as FLZ, KZL [9] and AFB against *C. albicans* [10] have also been documented, as well as combining azoles with *Mentha piperita* essential oils against *Candida* spp., *Cryptococcus neoformans* and *Trichophyton mentagrophytes* [15].

Azoles act against C14α-demethylase in biosynthesizing ergosterol, blocking its synthesis and causing toxic sterols to accumulate, interrupting the function of ergosterol in the cell membrane [36]. However, one of the mechanisms of resistance to azoles expressed by these pathogens is a reduction in intracellular azole accumulation; this may occur owing to a lack of drug penetration due to low levels of ergosterol or a possible decrease in the ratio between phosphatidylcholine and phosphatidylethanolamine in the plasma membrane, which can change the barrier function of the membrane [37]. In previous works, we demonstrated the damage to the membrane permeability of these yeasts as one of the mechanisms of ISO’s antifungal action against *C. tropicalis*. This damage was attributed to the inhibition of ergosterol synthesis [25,27], so we suggest that the synergistic effect of ISO with FLZ involves substantially permeabilizing the cell membrane, enabling the greater intracellular accumulation of ergosterol pathway intermediate sterols in addition to more ISO and FLZ entry, creating the potential for substantial toxic effects using low doses of ISO and FLZ. However, further studies are required to elucidate the synergy mechanism of ISO in combination with FLZ, AFB and CASP.

On the other hand, in this study, we found that orally administering different doses of the monoterpene ISO to female mice (*Mus musculus*) of the Balb/c lineage did not have any significant toxic effects that can be attributed to ISO treatment. It does not alter the weight of the animals in the acute phase or the chronic phase. According to the analyses performed on the blood, BAL and PCL, only the number of eosinophils increased, mainly in the chronic phase, which must be related to time and constant exposure to the compound over 98 days.

Malignant or premalignant lesions at the tissue level can cause immune system cells to infiltrate and, together with the tissue’s own cells, initiate an immune response at the local level that can be the Th1 type (interleukin-2 (IL-2), tumor necrosis factor-alpha (TNF-α) and interferon-gamma (IFN-γ)) of the Th2 type (IL-4, IL-5, IL-6, IL-9, IL-10 and IL-13) or both [38]. We determined the patterns of IFN-γ and TNF cytokines in the plasma, BAL and PCL of each animal treated with ISO. The results did not show statistically significant values when G3, G4, G5 and G6 (treated with ISO) were compared with the controls G1 and G2 (treated with water and oil, respectively). Likewise, in the chronic phase, no positive levels of any cytokine (in pg/mL) were detected. There was no increase in the cytokines, suggesting no inflammation process in the blood of the animals analyzed.

Generally, the histopathological studies showed non-significant changes in the samples analyzed from the experimental groups: the kidney fragments showed few microscopic alterations; the lung fragments showed changes such as insignificant hemorrhage, inflammatory infiltrate and thickening of the alveolar septa. Most of the liver fragments presented few perivascular infiltrates, and pyogranulomatous granulomas were identified only in the oil control and chronic study at 200 mg. Generally, no significant signs of toxicity were observed. This was corroborated by the biochemical findings, revealing no increase or significant difference in the amounts of GOT or GPT in the mice to which ISO was administered compared with the control groups in the acute phase; liver enzymes were within the standard. In the chronic phase, there was a significant decrease in GPT levels in G4 and G5 compared with group G2; GOT levels were not detected since they were expressed when they were above the control group values. These findings demonstrate that no toxicity can be attributed to ISO in the doses administered. These results are consistent with reports [39] of in vivo experiments using two groups of hamsters (*Mesocricetus auratus*), where the toxicity of the monoterpenes carvacrol and thymol (100 mg/kg) was evaluated, indicating that they were safe and effective treatments with few side effects on the liver and that they were very promising candidates for developing effective drugs. However, despite the valuable biological activities of monoterpenes, some of them (α-terpinene, camphor, citral, limonene, pulegone and thujone) have also shown a toxic character in in vitro and in vivo studies, so their use must be closely controlled [40].

Our results position ISO as an interesting compound to continue studying. In the future, it could be used as a therapeutic tool to control pathogenic microorganisms. It is important to conduct further research to validate these findings in humans.

## 4. Materials and Methods

### 4.1. Reagents

RPMI 1640 was obtained from Thermo Fisher Scientific (Waltham, MA, USA); 3-N-morpholinopropanesulfonic acid (MOPS) was obtained from Merck (Burlington, MA, USA); sabouraud dextrose agar (SDA), sabouraud dextrose broth (SDB), fluconazole (FLZ), amphotericin B (AFB), caspofungin (CASP) and 2,3,5-triphenyltetrazolium chloride (TCC) used in the synergism experiments were obtained from Sigma-Aldrich (São Paulo, SP, Brazil); xylazine and ketamine were obtained from Syntec (Diadema, SP, Brazil), and Vetnil (São Paulo, SP, Brazil); ethylenediamine tetra acetic acid (EDTA) was obtained from Dinamica (São Paulo, SP, Brazil); phosphate-buffered saline (PBS) citrate was obtained from LGC (São Paulo, SP, Brazil); Turk’s solution (3% acetic acid and 1% methylene blue) was obtained from Dinamica (São Paulo, SP, Brazil), and Synth (Diadema, SP, Brazil); rapid panoptic dye was obtained from Laborclin (São Paulo, SP, Brazil); buffered formalin was obtained from Synth (Diadema, SP, Brazil); hematoxylin–eosin (HE) was obtained from Easypath (São Paulo, SP, Brazil); TGO substrate was obtained from Labtest (São Paulo, SP, Brazil); TGP substrate solution was obtained from Labtest (São Paulo, SP, Brazil); NaOH was obtained from Neon (Suzano, SP, Brazil); blocking solution (1X PBS + 1% bovine serum albumin (BSA)) was obtained from Sigma-Aldrich (São Paulo, SP, Brazil); Tween 20 was obtained from Dinamica (São Paulo, SP, Brazil); sulfuric acid was obtained from Dinamica (São Paulo, SP, Brazil); standard recombinant cytokine, capture antibody, detection antibody (biotinylated secondary antibody—1: 250), detection antibody (biotinylated secondary antibody), streptavidin enzyme, TMB Solution A and Solution B (3,3′,5,5′-tetramethylbenzidine) and cytokines were obtained from BD Biosciences (San Diego, CA, USA).

### 4.2. Obtaining and Identification of Isoespintanol

ISO was obtained as a crystalline amorphous solid from a petroleum benzine extract of *O. xylopioides* leaves, and its structural identification was performed via GC-MS, ^1^H-NMR, ^13^C-NMR, DEPT, COSY ^1^H-^1^H, HMQC and HMBC. Information related to obtaining and identifying this ISO was reported in our previous study [25].

### 4.3. Strains

Seven clinical isolates of *C. tropicalis* (107, 98, 92, 84, 81, 74 and 03) were used in this study. The isolates were cultured from blood culture and tracheal aspirate samples from hospitalized patients at the Salud Social S.A.S. in the city of Sincelejo, Colombia. All microorganisms were identified via standard methods: Vitek 2 Compact, Biomerieux SA, YST Vitek 2 Card and AST-YS08 Vitek 2 Card (Ref 420739). SDA medium and BBL CHROMagar Candida medium were used to maintain the cultures until the tests were carried out.

### 4.4. Isoespintanol Action in Combination with Commercial Antifungals

To obtain the fractional inhibitory concentration indices (FICIs) of the ISO in combination with AFB, FLZ and CASP against *C. tropicalis*, we followed the methodologies proposed by Donadu [41]. Serial dilutions were made in RPMI 1640 broth with 0.1% 2,3,5-triphenyltetrazolium chloride (CTT), reaching final concentrations in a range of 1000—31.25 µg/mL, 4—0.125 µg/mL, 128—4 µg/mL and 1—0.031 µg/mL for ISO, AFB, FLZ and CASP, respectively. The assay was carried out in a total volume of 200 μL per well distributed as follows: 50 μL ISO + 50 µL of CASP, AFB and FLZ were added until the previously described concentrations were reached, as was 100 µL of fungal inoculum at a concentration of 10^6^ CFU/mL. Absorbance readings were measured immediately using a Chromate 4300 ELISA reader at a wavelength of 630 nm, and measurements were subsequently made after 24 h of incubation at 37 °C. The FICIs were calculated using the following equation [42]:$$\mathrm{FICIs}=\frac{{\;\mathrm{MIC}}_{90}\;\mathrm{ISO}\;\mathrm{in}\;\mathrm{combination}}{{\;\mathrm{MIC}}_{90}\;\mathrm{ISO}\;\mathrm{single}}+\frac{{\;\mathrm{MIC}}_{90}\;\mathrm{antifungal}\;\mathrm{in}\;\mathrm{combination}}{{\;\mathrm{MIC}}_{90}\;\mathrm{antifungal}\;\mathrm{single}\;}$$

The results were interpreted following the approach used in [43]. The FICIs are considered to have a synergistic effect (FIC index ≤ 1.0), a commutative effect (FIC index = 1), no interaction (1.0 < FIC index ≤ 2.0) or an antagonistic effect (FIC index > 2.0).

### 4.5. Preparation of Isoespintanol for Toxicity Experiments

The ISO crystals were dissolved in edible vegetable oil (used as a vehicle) at 45 °C to facilitate dilution. Subsequently, a stock solution was prepared at a concentration of 40 mg/mL with the help of a vortex mixer, and from this, the ISO dilutions were prepared at 25, 50, 100 and 200 µg/mL using the same vehicle.

### 4.6. Animals

The experimental design of this project was based on the recommendations of the Ethical Principles of Animal Experimentation adopted by the Brazilian Society of Laboratory Animal Science (SBCAL) and approved by the Committee on Ethics in the Use of Animals (CEUA) of the Federal University of São Carlos (UFSCar), under opinion n°. 4783100223. Female mice (*Mus musculus*) of the Balb/c lineage weighing between 15 and 18 g were used, supplied by ANILAB Animals De Laboratório Criação E Comercio Ltd. a. These animals had a specific pathogen free (SPF) certificate, guaranteeing that they were free of pathogens. All animals were kept in the vivarium of the Department of Morphology and Pathology of the Federal University of São Carlos (DMP—UFSCar), with free access to water and food for rodents in individual cages with air control (ALESCO).

### 4.7. Exposure to Isoespintanol and Experimental Groups

As shown in the experiment design (Figure 10), six experimental groups were established; all animals were weighed (between 15 and 18 g) and randomly distributed. Each group received oral treatment with a final volume of 100 μL. In the acute phase experiments, 55 animals were used (10 for each ISO group, 7 for the water control and 8 for the oil control); the treatment was carried out every 14 days, occurring on days 0 and 14. In the chronic phase, 55 animals were used (10 for each ISO group, 7 for the water control and 8 for the oil control); the treatment was carried out on days 0, 14, 28, 42, 56, 70 and 84.

Table 4 shows the animals’ exposure to the treatments in each experimental group, carried out in the acute phase after inoculation for 14 days and in the chronic phase for 90 days.

### 4.8. Animal Weight Measurement

Animals were weighed to check for changes in body mass and behavior after exposure to the compound. Weighing was always carried out in the morning and on days 0 and 14 to study the acute phase. To study the chronic phase, they were weighed on days 0, 14, 28, 42, 56, 70 and 84.

### 4.9. Euthanasia, Blood Cell Collection and Counting, PCL and BAL

Mice were sacrificed with xylazine and ketamine at 20 mg/kg intra-peritoneally (ip) on day 15 (acute phase) and day 90 (chronic phase). The blood was obtained by puncturing the left brachial vein, using EDTA as an anticoagulant at a final concentration of 0.3 M. To obtain the BAL, 2 mL of citrated PBS (1X [phosphate-buffered saline solution: 8 g of NaCl, 0.2 g of KCl, 1.15 g of Na_2_HPO_4_, 0.2 g of KH_2_PO_4_ and 1 L of distilled water] + 0.5% sodium citrate) was injected through a catheter into the tracheas of the animals. To obtain the PCL, 3 mL of citrated PBS was used, introducing a needle into the peritoneal cavity, from which PCL was then extracted. Global blood cell counting, BAL and PCL were performed individually in a Neubauer chamber, and samples were added to Turk’s solution (3% acetic acid and 1% methylene blue) at a dilution of 1:20. For differential count and percentages of cells (eosinophils, neutrophils and mononucleated leukocytes), blood smears and PCL slides were prepared in a Citospin centrifuge (Serocyte^®^ model 2400, Thermo Scientific™, Hampton, NH, USA), stained with rapid panoptic dye. On each slide, 100 cells were counted via optical microscopy with a final magnitude of 1000. The plasma was subsequently stored at −20 °C to measure cytokines and transaminases GOT and GPT.

### 4.10. Histological Evaluation of Lungs, Livers and Kidneys

To analyze the possible involvement of tissues and organs due to indirect toxicity, lungs, livers and left kidneys from animals in both the acute and chronic phases were extracted and stored, standardizing animals 1 and 2 of each group. First, an approximately 4 cm surgical incision was made longitudinally just below the rib cage of each mouse. A small incision was then made in the diaphragm to expose the sternum bone. Through this incision, the thoracic bone cavity was removed for direct access and lung collection. When removing the organs from the abdominal cavity, the same incision was used. Therefore, the liver and kidneys were removed sequentially. After removal, the organs were washed with 1X PBS solution and dried on paper. Subsequently, fixation was carried out in buffered formalin (4 g of NaH_2_PO_4_, 6.5 g of Na_2_HPO_4_, 100 mL of formaldehyde and 900 mL of distilled water). Subsequently, they were sent to undergo the previously established specific processes to prepare the histological slides, which were stained with hematoxylin–eosin. All analysis of the fragments was performed using an Opticam Binocular Optical Microscope (model O400S, São Paulo, Brazil), using a 40× objective.

### 4.11. GOT and GPT Dosage

The test was performed according to the manufacturer’s instructions (Labtest). Initially, the test tubes were identified, and 100 µL of GOT substrate or GPT substrate solution was added, which was then incubated in a water bath at 37 °C for 2 min. Then, 20 µL of blood plasma from animals exposed or not exposed to the compound was added, and another incubation was carried out in a water bath (37 °C) for 60 min for the GOT test and 30 min for the GPT test. After incubation, 100 µL of color reagent (AST/GOT and ALT/GPT Liquiform, Labtest, Lagoa Santa, MG, Brazil) was added. After 20 min at room temperature, 1 mL of NaOH solution (1.25 mol/L) was added, and we waited 5 min. Finally, 300 µL of the samples (in triplicate) was transferred to 96-well microtiter plates, and the absorbance reading was performed at a wavelength of 505 nm using a plate spectrophotometer (Thermo Scientific™ Multiskan™ GO Microplate Spectrophotometer, Waltham, MA USA). In parallel with the sample preparation, the calibration curve was prepared (according to the kit) for subsequent calculations of the GOT and GPT levels in the analyzed samples.

### 4.12. Measurement of IFN-γ and TNF Cytokines in Plasma, BAL and PCL

To measure IFN-γ and TN, with a direct ELISA (OptEIA^TM^ Kit, BD Biosciences, San Jose, CA, USA), 96-well high-affinity microtiter plates were used according to the protocol described below. Between each step, the wells were washed with 300 μL of washing solution (1X PBS [phosphate buffer saline: 8 g NaCl, 0.2 g KCl, 1.15 g Na_2_HPO_4_, 0.2 g KH_2_PO_4_ and 1 L distilled water] + 0.05% Tween 20, pH 7.4).

Initially, the plate was sensitized with 100 µL/well of a solution containing the capture antibody (1:250) diluted in 0.1 M of carbonate buffer—pH 9.5 (7.13 g of NaHCO_3_ and 1.59 g of Na_2_CO_3_ in 1 L of milliwater). Plates were incubated for up to 18 h at 4 °C. After this period, the supernatant was discarded, the plate was washed and non-specific binding sites were blocked by adding 200 µL of blocking solution (1X PBS + 1% bovine serum albumin [BSA]—Sigma). Again, the plates were incubated for 1 h at room temperature and washed. Subsequently, different dilutions of the standard recombinant cytokine (curve) and the samples under study (50 µL/well) were added in triplicate. After a 2 h incubation, a new washing cycle was performed; 100 µL/well of detection antibody (biotinylated secondary antibody, 1:250) was added to the TNF, and 100 µL/well of detection antibody (biotinyl-sided secondary antibody 1:250) was added with enzyme (1:250) to the IFN-γ. After a new incubation of 1 h and 30 min at room temperature and a new washing cycle, 100 µL/well of the enzyme streptavidin (1:250) was added to the TNF, and after 30 min, the mixture was washed again, and 100 µL/well of substrate was added. For IFN-γ, after 1 h and 30 min of secondary antibody + enzyme incubation, the plate was washed again, and 100 µL/well of substrate was added. The substrate was a 1:1 mixture of TMB Solution A and Solution B (3,3′,5,5′-tetramethylbenzidine). Finally, the reaction was blocked by adding 50 µL/well of 2 M sulfuric acid (H_2_SO_4_). Absorbance was read at a wavelength of 450 nm using a plate spectrophotometer (Thermo Scientific™ Multiskan™ GO Microplate Spectrophotometer, Waltham, MA USA), with concentrations calculated from the titration curve of the cytokine standards. The final concentrations were expressed in pg/mL.

### 4.13. Statistical Analysis

The data obtained in this study were analyzed using GraphPad Prism 7.0 (San Diego, CA, USA). The entire study was carried out at least sixfold, with the n value of the sample varying from 6 to 10 (N = 6–10) in independent experiments. Discrepant data were identified using Grubbs analysis, followed by the Shapiro–Wilk test to verify the parametric or non-parametric nature of the data. To achieve this, the ANOVA test (analysis of variance) and Tukey’s multiple comparisons post-test were applied to the parametric data (the results are presented as the mean and standard deviation). For non-parametric data, the Kruskal–Wallis test and Dunn’s multiple comparison post-test were used (the results are presented as the median with the upper and lower quartiles: Me [Q1; Q3]). Statistical significance was established at *p* < 0.05; the results are expressed as mean ± SEM, except for the cytokine doses (TNF and IFN-γ)—which are expressed as mean ± SD owing to the analysis carried out with the plasma set of animals in each group—and analyzed using GraphPad Prism, version 9 (San Diego, CA, USA). For the analysis, the one-way ANOVA test (one-way analysis of variance) was used, and the post-test was performed using Tukey’s method (Tukey’s multiple comparisons test). The significance level adopted was 5%, where *p* ≤ 0.05.

## 5. Conclusions

Herein, we report a synergistic effect of ISO in combination with the antifungals FLZ, CASP and AFB, showing a greater effect in the combined ISO-FLZ treatment compared with the ISO-CASP and ISO-AFB treatments in this study. Furthermore, the side effects of ISO in female mice (*Mus musculus*) of the Balb/c lineage were not significant, indicating that ISO does not exert toxic effects at the concentrations tested on these mice. These results can serve as references for the continued study of this monoterpene, given the important antimicrobial potential previously reported, as an interesting tool for treating and controlling drug-resistant in-hospital pathogens.

## Figures and Tables

**Figure 1 molecules-29-04417-f001:**
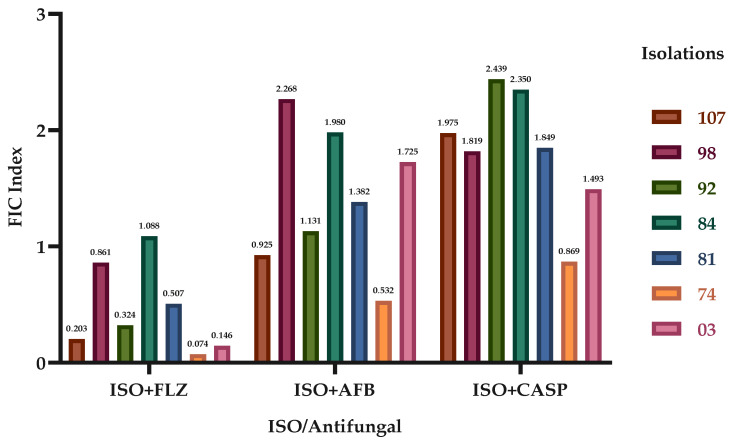
Effect of ISO in combination with FLZ, AFB and CASP against *C. tropicalis*. The fractional inhibitory concentration indices (FICIs) of the different clinical isolates of *C. tropicalis* are shown.

**Figure 2 molecules-29-04417-f002:**
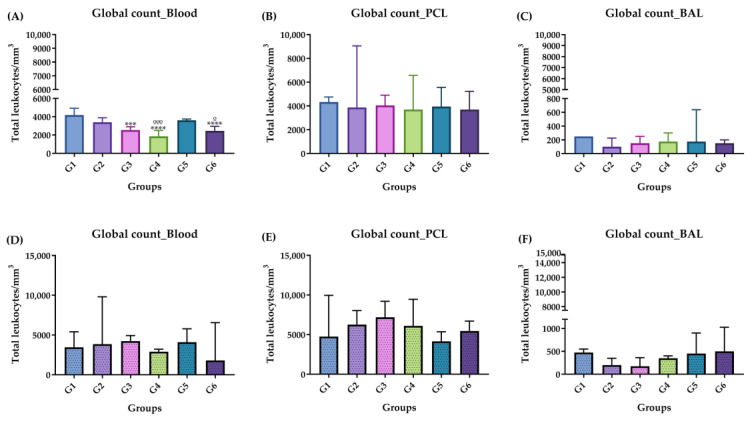
Global leukocyte count, acute phase (**A**–**C**) and chronic phase (**D**–**F**). (**A**) Acute phase: blood; (**B**) acute phase: PCL; (**C**) acute phase: BAL; (**D**) chronic phase: blood; (**E**) chronic phase: PCL; (**F**) chronic phase: BAL. Groups: G1: CTRL + water; G2: CTRL + oil; G3–G6: 25, 50, 100 and 200 mg/mL of ISO at 14 days. (*) vs. G1: *** *p* ≤ 0.001; **** *p* ≤ 0.0001. (°) vs. G2: ° *p* ≤ 0.05, °°° *p* ≤ 0.0001. The results are presented with the mean and standard deviation (**A**,**E**). The results are presented as the median with the upper and lower quartiles: Me [Q1; Q3] (**B**–**D**,**F**).

**Figure 3 molecules-29-04417-f003:**
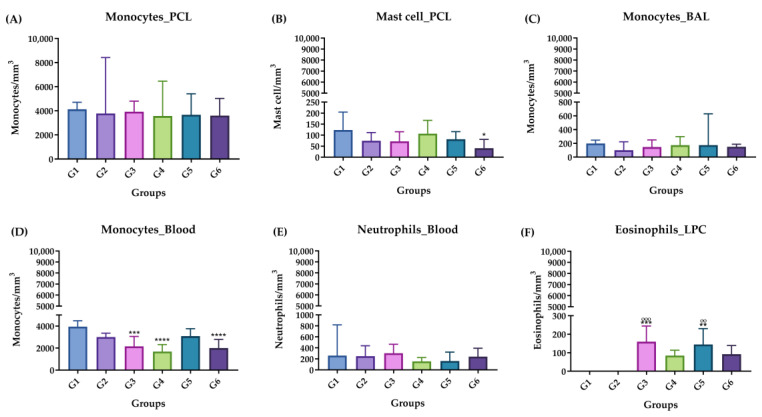
Differential leukocyte count, acute phase. (**A**) Monocytes in PCL. (**B**) Mast cells in PCL. (**C**) Monocytes in BAL. (**D**) Monocytes in blood. (**E**) Neutrophils in blood. (**F**) Eosinophils in blood. Groups: G1: CTRL + water; G2: CTRL + oil; G3–G6: 25, 50, 100 and 200 mg/mL of ISO after 14 days. (*) vs. G1: * *p* ≤ 0.05; ** *p* ≤ 0.01; *** *p* ≤ 0.001; **** *p* ≤ 0.0001. (°) vs. G2: °° *p* ≤ 0.01; °°° *p* ≤ 0.001. The results are presented with the mean and standard deviation (**B**,**D**). The results are presented as the median with the upper and lower quartiles: Me [Q1; P3] (**A**,**C**,**E**,**F**).

**Figure 4 molecules-29-04417-f004:**
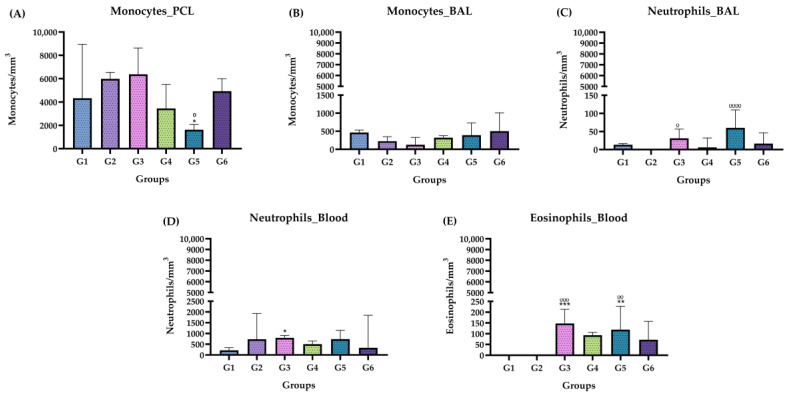
Differential leukocyte count, chronic phase. (**A**) Monocytes in PCL. (**B**) Monocytes in BAL (**C**) Neutrophils in BAL. (**D**) Neutrophils in blood. (**E**) Eosinophils in the blood. Groups: G1: CTRL + water; G2: CTRL + oil; G3–G6: 25, 50, 100 and 200 mg/mL of the ISO at 90 days. (*) vs. G1: * *p* ≤ 0.05; ** *p* ≤ 0.01; *** *p* ≤ 0.001. (°) vs. G2: ° *p* ≤ 0.05; °° *p* ≤ 0.01; °°° *p* ≤ 0.001, °°°° *p* ≤ 0.0001. The results are presented with the mean and standard deviation (**B**,**D**). The results are presented as the median with the upper and lower quartiles: Me [Q1; P3] (**A**–**E**).

**Figure 5 molecules-29-04417-f005:**
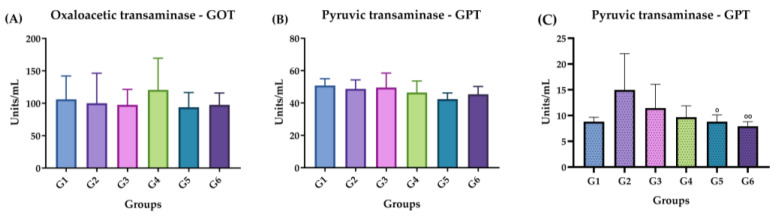
GOT and GPT doses in the acute and chronic phases. (**A**) Acute phase: GOT. (**B**) Acute phase: GPT. (**C**) Chronic phase: GPT. Groups: G1: CTRL + water; G2: CTRL + oil; G3–G6: 25, 50, 100 and 200 mg/mL of ISO after 14 days. (*) vs. G1: (°) vs. G2: ° *p* ≤ 0.05; °° *p* ≤ 0.01. The results are presented with the mean and standard deviation.

**Figure 6 molecules-29-04417-f006:**
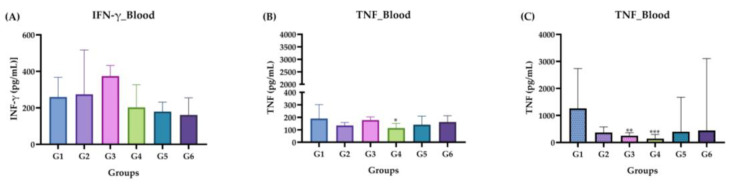
Measurement of INF-γ and TNF cytokine levels in the acute and chronic phase. (**A**) Acute phase: IFN-γ. (**B**) Acute phase: TNF. (**C**) Chronic phase: IFN-γ. Groups: G1: CTRL + water; G2: CTRL + oil; G3–G6: 25, 50, 100 and 200 mg/mL of ISO after 14 days. The results are presented as the median with the upper and lower quartiles: Me [Q1; P3]. (*) vs. G1: * *p* ≤ 0.05; ** *p* ≤ 0.01; *** *p* ≤ 0.001.

**Figure 7 molecules-29-04417-f007:**
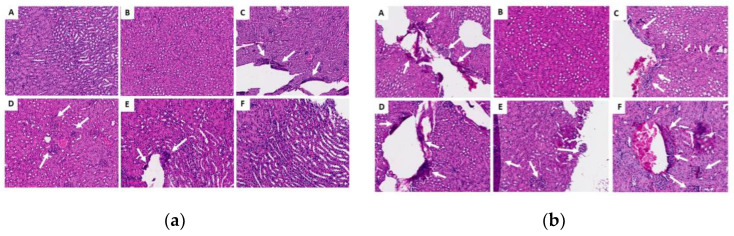
Histological sections of the kidneys in the acute (**a**) and chronic (**b**) phases. (A) Control G1 (water); (B) control G2 (oil); (C) G3 (25 mg/kg); (D) G4 (50 mg/kg); (E) G5 (100 mg/kg) and (F) G6 (200 mg/kg). The white arrows indicate areas of lymphoplasmacytic infiltrates; there were no changes in the groups represented by images (A), (B) or (F) in the chronic phase.

**Figure 8 molecules-29-04417-f008:**
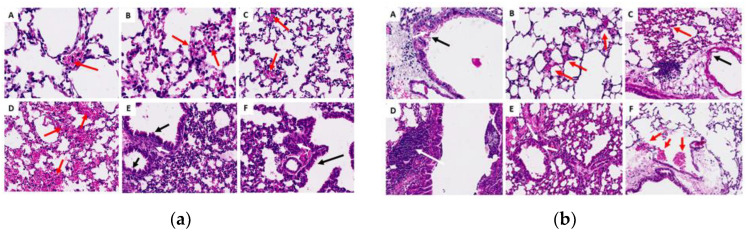
Histological sections of the lungs in the acute (**a**) and chronic (**b**) phases. (A) Control G1 (water); (B) control G2 (oil); (C) G3 (25 mg/kg); (D) G4 (50 mg/kg); (E) G5 (100 mg/kg) and (F) G6 (200 mg/kg). Red arrows indicate foci of hemorrhage; white arrows show areas of lymphoplasmacytic infiltrates; and black arrows represent thickening in the alveolar septa.

**Figure 9 molecules-29-04417-f009:**
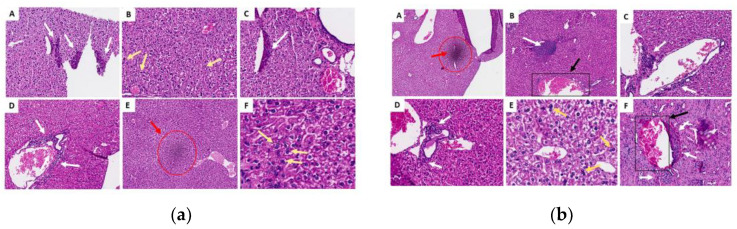
Histological sections of the liver in the acute (**a**) and chronic (**b**) phases. (A) Control G1 (water); (B) control G2 (oil); (C) G3 (25 mg/kg); (D) G4 (50 mg/kg); (E) G5 (100 mg/kg) and (F) G6 (200 mg/kg). White arrows indicate areas of lymphoplasmacytic infiltrates; yellow arrows indicate the presence of binucleated hepatocytes (a sign of regeneration); red marks refer to discrete foci of necrosis and black marks represent areas of granuloma.

**Figure 10 molecules-29-04417-f010:**
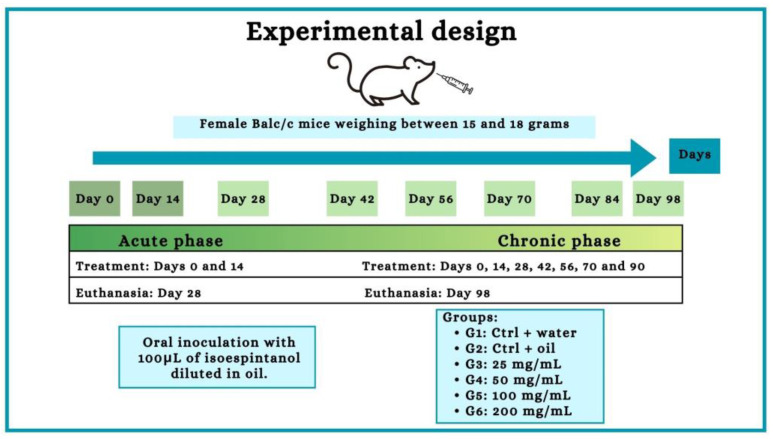
Experiment design. Female Balb/c mice received oral treatment with 100 μL of different concentrations of the ISO (25, 50, 100 and 2000 mg/mL) every 14 days in an acute (28 days) and chronic (98 days) phase study.

**Table 1 molecules-29-04417-t001:** MIC_90_ values, individually and in combination, of ISO, FLZ, AFB and CASP against *C. tropicalis*.

*C. tropicalis*	MIC_90_ Single				MIC_90_ in Combination			FIC Indices			Effect		
	FLZ	AFB	CASP	ISO	ISO-FLZ	ISO-AFB	ISO-CASP	ISO-FLZ	ISO-AFB	ISO-CASP	ISO-FLZ	ISO-AFB	ISO-CASP
107	75.4	2.0	0.5	362.1	45.5–5.8	195.3–0.8	410.5–0.4	0.2	0.9	2.0	Sng	Sng	S.I.
98	8.0	2.1	0.5	304.1	44.8–5.7	439.4–1.8	339.2–0.3	0.9	2.3	1.8	Sng	Ant	S.I.
92	66.8	1.7	0.7	222.1	50.5–6.5	165.4–0.7	404.1–0.4	0.3	1.1	2.4	Sng	S.I.	Ant
84	358.3	2.4	0.5	416.0	394.1–50.5	483.3–1.9	511.9–0.5	1.1	2.0	2.4	S.I.	S.I.	Ant
81	529.3	1.6	0.5	299.8	141.7–18.1	237.6–1.0	341.9–0.3	0.5	1.4	1.8	Sng	S.I.	S.I.
74	428.5	2.5	0.5	360.1	24.0–3.1	121.4–0.5	175.3–0.2	0.1	0.5	0.9	Sng	Sng	Sng
03	410.0	2.0	0.6	332.7	44.1–5.6	347.5–1.4	316.6–0.3	0.1	1.7	1.5	Sng	S.I.	S.I.

Ant: antagonism; Sng: synergism; S.I.: no interaction.

**Table 2 molecules-29-04417-t002:** Weight of the animals during the acute phase. Days of ISO inoculation vs. average weight in grams of the animal group.

Days/Groups	G1	G2	G3	G4	G5	G6
0	20.88	20.48	22.36	20.5	20.8	21.36
14	22.12	22.09	22.42	21.34	22.12	21.33

**Table 3 molecules-29-04417-t003:** Weight of animals in the chronic phase. Days of ISO inoculation vs. average weight in grams of the animal group.

Days/Groups	G1	G2	G3	G4	G5	G6
0	21.39	20.6	21.16	19.94	21.45	22.85
14	21.99	21.66	22.79	21.05	23.18	23.66
28	22.72	22.45	23.58	21.52	24.31	24.08
42	22.51	22.91	23.51	21.81	24.66	23.77
56	23.21	23.19	24.70	21.99	25.10	24.37
70	24.13	23.47	24.87	21.79	25.35	24.83
84	24.09	21.76	25.17	22.67	25.71	25.11

**Table 4 molecules-29-04417-t004:** Exposure of animals to ISO carried out in the acute phase after inoculation for 14 days and in the chronic phase for 90 days.

Experimental Group	Route of Administration	Exposure	Volume/Animal
G1: CTRL + water	Oral	Water	100 µL
G2: CTRL + oil	Oral	Oil	100 µL
G3:25 mg/mL	Oral	ISO 25 mg/mL	100 µL
G4: 50 mg/mL	Oral	ISO 50 mg/mL	100 µL
G5: 100 mg/mL	Oral	ISO 100 mg/mL	100 µL
G6: 200 mg/mL	Oral	ISO 200 mg/mL	100 µL

## Data Availability

The data presented in this study are available in the article and the Appendix A.

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
