# Peer review of "Synergistic Antifungal Effect and In Vivo Toxicity of a Monoterpene Isoespintanol Obtained from Oxandra xylopioides Diels"

_molecules, 2024, doi:10.3390/molecules29184417_

Round 1

Reviewer 1 Report

Comments and Suggestions for Authors

The main object of this study was to assess the potential synergy of isoespintanol in combination with the antifungals, fluconazole, amphotericin B and cas- 21 pofungin, against clinical isolates of C. tropicalis. Antifungal drugs are relatively understudied in the field of infection control and as such it is encouraging to see new studies emerging, especially those which explore the potential synergy with other compounds. As the authors have mentioned synergistic combinations of compounds are not just effective in healing infections but can also reduce the emergence of antimicrobial resistance which has become a growing concern in recent years.

That being said, the authors of the manuscript present a very compelling case for the use of isoespintanol as a synergistic additive to well known antifungals. In addition I think the manuscript is very well presented and has covered most of the appropriate tests to reach their conclusions.

My recommendation would be to publish this manuscript with minor corrections.

There are one or two small typographic errors in the manuscript that I have mentioned below.

Line 16.  Abstract. The use of ‘they’ is redundant, please remove.

Line 17-18. Please rephrase for clarity.

Line 26. Do the authors mean ‘observation’ here instead of vision ?

Line 106. I am not too sure about MDPI molecules format but many journals insist that legends for figures are stand-alone. This would mean in this instance that abbreviations have to be spelled out.

Line 168. Figure 4. I understand that some of the y-axis are staggered but could the authors perhaps make the top end of these axis look less cluttered, i.e. less numbers ?

Line 183. Figure 5. Could the authors use consistent graphics on their diagrams ? For example 5C has black outline on bars, while 5A and 5B do not.

Line 195. Figure 6. Is there a square bracket missing at the end of the legend ?

Figure 10. ‘Design’ spelled with an additional e in the figure !

Comments on the Quality of English Language

First two lines of the abstract need brushing up for clarity and impact !

I have noted other typos in my comments above

Reviewer 2 Report

Comments and Suggestions for Authors

General Comments:

The manuscript titled "Synergistic antifungal effect and in vivo toxicity of the monoterpene isoespintanol obtained from Oxandra xylopioides Diels" presents interesting and significant findings on the antifungal potential and safety profile of isoespintanol when combined with conventional antifungal drugs. The study is well-structured and provides substantial data on both the synergistic effects and the in vivo toxicity of isoespintanol, making it a valuable contribution to the field of natural product-based antifungal therapies.

Strengths:

  1. Scientific Relevance: The study addresses a critical need for new antifungal strategies, particularly against drug-resistant Candida species, which are a growing concern in clinical settings.

  2. Experimental Design: The authors have employed a comprehensive set of methodologies to assess the synergistic effects of isoespintanol with fluconazole, amphotericin B, and caspofungin, as well as the in vivo toxicity in mice. The use of multiple clinical isolates and the detailed toxicity studies add robustness to the findings.

  3. Data Presentation: The figures and tables are clear and effectively present the data. The results are discussed in a logical manner, correlating the findings with previous studies in the literature.

Areas for Improvement:

  1. Mechanistic Insights: While the study effectively demonstrates the synergistic effects and lack of significant toxicity of isoespintanol, the mechanistic basis for the observed synergy, particularly at the molecular level, could be explored further. The discussion would benefit from a deeper analysis of how isoespintanol interacts with antifungal agents at the cellular level.

  2. Statistical Analysis: Although the statistical analyses are appropriately applied, some results in the tables and figures could benefit from additional clarification, such as specifying whether the presented p-values correspond to comparisons between specific groups (e.g., control vs. treated).

  3. Supplementary Materials: The authors mention supplementary materials, but these are not included in the provided document. It would be useful to review these materials to assess their contribution to the overall findings.

Comments on the Quality of English Language
  1. While the manuscript is generally well-written, there are minor grammatical errors and awkward phrasing in some sections. A thorough proofreading or professional editing could enhance the readability of the text.
